# Beyond the first glance: How human presence enhances visual entropy and promotes spatial learning

Tracy Sánchez Pacheco [1]*, Debora Nolte[1], Sabine U. König[1],
Gordon Pipa[1], Peter König[1,2]

1 Institute of Cognitive Science, University of Osnabrück, Osnabrück, Germany, 2 Department of Neurophysiology and Pathophysiology, University Medical Center Hamburg-Eppendorf, Hamburg, Germany

☯ These authors contributed equally and share senior authorship.
* tracysanchez@uos.de

## Abstract

Spatial learning emerges not only from static environmental cues but also from the social and semantic context embedded in our surroundings. This study investigates how human agents influence visual exploration and spatial knowledge acquisition in a controlled Virtual Reality (VR) environment, focusing on the role of contextual congruency. Participants freely explored a 1 km² virtual city while their eye movements were recorded. Agents were visually identical across conditions but placed in locations that were either congruent, incongruent, or neutral with respect to the surrounding environment. Using Bayesian hierarchical modeling, we found that incongruent agents elicited longer fixations and higher gaze transition entropy (GTE), a measure of scanning variability. Crucially, GTE emerged as the strongest predictor of spatial recall accuracy. A counterfactual mediation analysis indicated a small but reliable pathway via GTE and, for incongruent agents, a larger direct component not captured by GTE. These findings suggest that human-contextual incongruence promotes more flexible and distributed visual exploration, thereby enhancing spatial learning. By showing that human agents shape not only where we look but how we explore and encode space, this study contributes to a growing understanding of how social meaning guides attention and supports navigation.

## Author summary

When people explore a new environment, such as an unfamiliar city, they rely on what they see to understand and remember the space. Traditionally, research has focused on stable features like buildings or landmarks. However, real-world environments also include people, whose presence can shape how we explore and learn. In this study, participants explored a virtual city while their eye movements

**Data availability statement:** All data and analysis code for this study are publicly available. The complete dataset and supplementary materials are hosted on the Open Science Framework (OSF) and can be accessed via DOI: 10.17605/OSF.IO/3VNMJ. The Wiki, of the OSF project now states how to interact with the GitHub repo. The code used to calculate entropy measures is available on GitHub at https://github.com/tracysanchez/Human_Agents_Impact_Nav, specifically under the `Exp1/Entropy/` and `Exp2/Entropy/` directories.

**Funding:** The author(s) declare that financial support was received for the research and/or publication of this article. TS and GP received support from the Deutscher Akademischer Austauschdienst (DAAD), Grant No. 57440921 (https://www.daad.de/en/). Additionally, PK received funding from GRK 2340: Computational Cognition. The funders had no role in study design, data collection and analysis, decision to publish, or preparation of the manuscript.

**Competing interests:** The authors have declared that no competing interests exist.

were tracked. Some human figures matched their surroundings, while others appeared out of place. We found that people looked longer at those unexpected figures and that their gaze patterns became more flexible and varied afterward. This broader visual exploration helped them remember the layout of the city more accurately. Our results suggest that human presence, especially when it disrupts expectations, can promote more effective learning by encouraging more complex visual engagement. These findings can provide insight into how strategically placed social cues enhance attention and memory, and more importantly, how they influence our patterns of visual exploration.

## Introduction

Spatial cognition has traditionally been studied through static environmental cues, such as landmarks and architectural layout [1]. These elements provide stable reference points that aid wayfinding and memory encoding. However, real-world spatial cognition extends beyond static structures. Environments contain dynamic elements, here, dynamic denotes contextual/semantic variability (i.e., elements whose presence can change) and does not imply physical motion, including socially relevant cues, that guide spatial understanding. Among these, humans serve as particularly meaningful reference points, as their presence and placement within an environment convey implicit information about its function, affordances, and social relevance [2]. Unlike fixed landmarks, human agents contextualize the environment by embedding scenes with social and semantic meaning. Depending on their relevance to the setting, their presence may also interfere with spatial learning through social cueing [3]. Despite their ubiquity in real-world navigation, the role of fellow humans' contextual relevance in shaping spatial knowledge formation remains underexplored.

In virtual environments, recent findings suggest that human agents in the form of social wayfinding cues can enhance spatial awareness and knowledge acquisition [4]. Sánchez-Pacheco et al. (2025) examined how the type and contextual relevance of virtual agents shaped spatial cognition, showing that although the mere presence of agents produced only local differences in exploration behaviour, agent characteristics such as semantic congruence or incongruence with their surroundings meaningfully modulated visual attention and recall [5]. Their work emphasizes that human agents contribute semantic relevance to an environment by shaping how spatial relationships are perceived. For instance, a construction worker positioned in front of a construction site reinforces the semantic association of that location. In contrast, if the same worker stands in front of a basketball court, the mismatch creates an incongruence that enhances participants' ability to remember the location. These findings highlight two complementary mechanisms: (a) human agents provide semantic structure by signalling how locations relate to plausible human activities, and (b) semantically incongruent agents can amplify spatial distinctiveness, thereby promoting deeper encoding. Yet how this contextual relevance translates into changes in perception and encoding remains to be understood.

A key starting point is vision, which plays a central role in how spatial information is perceived, organized, and remembered. While navigating an environment, individuals direct their visual exploration toward areas of interest before determining goal-oriented body movement [6]. Visual exploration involves a balance between exploitation, where attention is concentrated on task-relevant, familiar elements, and exploration, where fixations shift flexibly to gather new spatial information [7]. Fixations have traditionally been considered indicators of exploitation, as they reflect the selection of known or relevant information. In contrast, transitions between fixations encode the temporal and sequential structure of exploration, capturing how attention is dynamically allocated across a scene [8]. The interplay of visual exploitation and exploration is particularly relevant in understanding how human agents shape visual behavior. To study this balance, we need a quantitative description of how gaze sampling is organized over time. In what follows, we quantify this sequence-dependent organization using gaze transition entropy.

To capture the sequence-dependent organization of visual sampling, entropy provides two complementary lenses. Stationary gaze entropy summarizes how widely fixations are distributed across regions of interest while ignoring temporal order. By contrast, gaze transition entropy (GTE) is computed from empirical fixation–transition matrices and quantifies the conditional uncertainty of the next fixation given the current one [9]. Grounded in information theory [10], the two measures answer different questions: stationary entropy describes how evenly gaze tends to accumulate in areas of interest [11], whereas GTE characterizes how sampling unfolds conditional on the current fixation category [12]. Because our aims concern rapid, agent-anchored changes in sampling, we focus on GTE. In this framing, higher GTE indicates greater conditional uncertainty about the next fixation, that is, a more diversified set of next-step choices. Accordingly, we use GTE to index how viewing policies reorganize in real time, setting up the link between entropy and goal-directed navigation.

We treat GTE as a proxy for visual sampling, tracking how fixations on human agents shape subsequent fixations. We posit that prediction violations broaden sampling and raise GTE, whereas concentrated evidence narrows sampling and lowers GTE. Because broader sampling is costly [13], it likely signals high expected information gain and may enhance recall. Consistent with this account, entropy-based measures track attention [14], rise with task difficulty [15], and align with perceived proficiency [16]. What remains unknown, and motivates this study, is how agent–context congruence modulates GTE and whether these changes affect later spatial recall.

To quantify fixation dynamics, gaze transition entropy (GTE) provides a structured measure of dispersed versus predictable visual exploration. Higher GTE values indicate a more exploratory scanning strategy, while lower values reflect a focused, goal-directed search. Beyond capturing fixation dispersion, GTE has been linked to higher-order cognitive processes, including attentional allocation [14], task difficulty [15], and perceived task proficiency [16]. Social presence has also been shown to influence fixation behavior, either by clustering fixations due to social salience [17] or by modulating them based on contextual relevance [18]. However, it remains unclear how social context, particularly the congruence between agents and their surroundings, shapes variability in visual exploration as captured by GTE.

To address this question, we designed a 1 km² virtual city composed of 236 buildings and 56 human agents. A subset of agents held objects implying specific actions (e.g., a shovel or a doughnut) and were positioned either in semantically congruent contexts (e.g., a construction worker at a construction site) or incongruent ones (e.g., a construction worker in front of a doughnut shop). Participants freely explored the city while their gaze behavior was recorded, allowing for the analysis of fixation durations and GTE in response to different contextual conditions.

Here, we investigate how the presence of human agents influences visual exploration and spatial knowledge acquisition in virtual environments. Specifically, we examine whether action-implying agents modulate attention allocation and encoding processes depending on the congruence between agent and environment. Relative to [5], we introduce gaze transition entropy (GTE) as an information-theoretic index of exploration policy, quantify its pre/post dynamics in 30-s windows time-locked to agent fixations, and model pointing accuracy with a Bayesian hierarchical Gamma regression that includes GTE and z-scored dwelling time covariates under explicit treatment contrasts. As a last step, we further estimate direct and mediated effects using counterfactual mediation. We hypothesize that human agents encourage broader

exploration by increasing attentional shifts, and that semantic congruence drives both visual behavior and subsequent spatial learning.

## Materials and methods

### Ethics statement

This study was approved by the Ethics Committee of the University of Osnabrück (approval number: Ethik56-2020), and written informed consent was obtained from all participants prior to participation.

### Participants

The dataset analyzed in this study was previously collected and described by Sánchez Pacheco et al. [5]. While the original analysis focused on spatial navigation and memory performance, the present study builds on this dataset by introducing a novel analysis of gaze behavior and fixation dynamics, offering new insight into the visual processes that accompany spatial learning in socially contextualized environments. Briefly, the study consisted of two experiments, for which we initially recruited 70 participants evenly distributed across both experiments. All participants had normal or corrected-to-normal vision. Due to attrition, the final sample comprised 53 participants. Ten participants were unable to complete the study due to illness or scheduling conflicts, three withdrew due to motion sickness, and four were excluded due to incomplete data. The final distribution included 27 participants in Experiment 1 (14 male, 13 female) and 26 in Experiment 2 (11 male, 15 female). In Experiment 1, participants ranged in age from 19 to 46 years ($M = 23.92$, $SD = 6.72$), while in Experiment 2, ages ranged from 19 to 31 years ($M = 22.29$, $SD = 2.90$).

### Experimental design and setup

Participants freely navigated a virtual city while their eye movements were continuously recorded to investigate how contextual agents influence visual exploration and spatial learning. The experimental setup, virtual environment, and procedure followed the methodology outlined in Sánchez-Pacheco et al. [5]. Participants explored a $1\,\text{km}^2$ virtual city with 236 buildings categorized into residential areas and public spaces, such as restaurants, retail stores, recreational areas, and construction sites (see Fig 1A). Among these, 52 buildings were marked with street art, evenly distributed across residential and public spaces, designating them as task-relevant locations. Four large peripheral buildings functioned as global landmarks, serving as reference points for navigation. This structured environment allowed for controlled manipulation of spatial cues, ensuring visual behavior could be analyzed in response to architectural elements and agent placement.

Each participant completed five 30-minute exploration sessions, followed by a final 60-minute test session to assess spatial knowledge. During all sessions, participants were free to explore the city while their eye movements were recorded using a VR-based eye-tracking system. The final session included pointing tasks to evaluate their internal spatial representations. To ensure data quality, eye-tracker calibration was repeated every 10 minutes. This protocol allowed for high-resolution tracking of gaze behavior across repeated exposures, forming the basis for linking visual exploration with spatial learning outcomes.

The exploration and task assessment sessions were conducted with a desktop computer with an Intel Xeon W-2133 CPU, 16 GB RAM, and an NVIDIA RTX 2080 Ti graphics card. The VR environment was rendered using an HTC Vive Pro Eye head-mounted display (HMD), operating at a refresh rate of 90 Hz with an effective field of view of approximately 110°. We used four SteamVR Base Stations 2.0, an HTC VIVE body tracker 2.0, and Valve Index controllers to monitor participants' positions within the environment. This combined setup achieved sub-millimeter precision in capturing the head, body, and eye positions, rotation, and orientation.

   

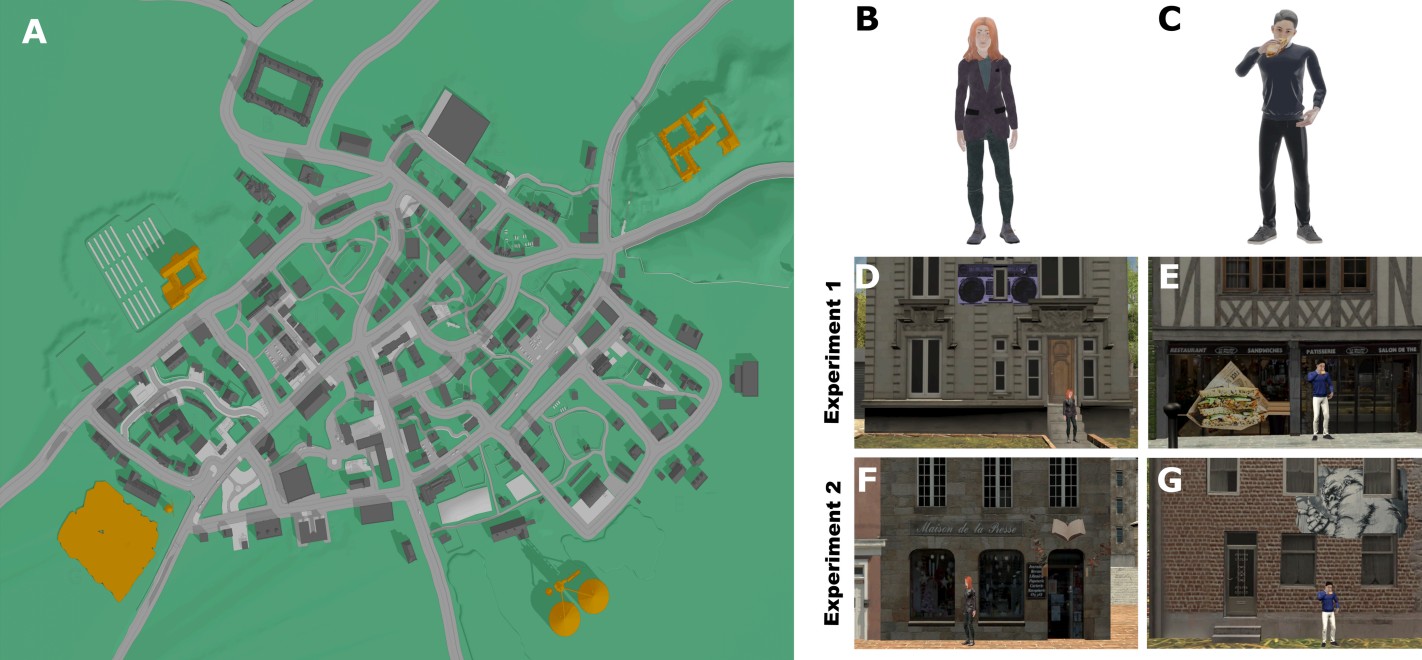

**Fig 1. Experimental design and virtual environment.** (A) Bird's-eye view of the 1 km² virtual city, showing buildings and paths. Orange marks indicate the four global landmarks. (B) Example of an acontextual agent. (C) Example of a contextual agent. (D–G) Agent placements across experimental conditions. In Experiment 1, (D) acontextual agents in residential areas, and (E) contextual agents in congruent locations (e.g., a sandwich in front of a sandwich shop). In Experiment 2, (F) acontextual agents in both residential and public spaces, and (G) contextual agents in incongruent settings.

The key experimental manipulation involved the placement of contextual agents (see Fig 1). Overall, we placed 56 agents throughout the virtual city, 28 contextual and 28 acontextual. In Experiment 1, acontextual agents were placed in neutral residential settings (Fig 1B, 1D) and contextual agents in congruent locations, where their held objects aligned with the surrounding environment, such as a sandwich in front of a sandwich shop (Fig 1C, 1E), reinforcing contextual consistency. In Experiment 2, all agents were redistributed: acontextual agents were placed evenly across residential and public spaces, the latter including locations associated with commercial and recreational activities (Fig 1F), while previously congruent agents were relocated to incongruent settings, ensuring they were placed in public or residential areas that did not match their held objects (Fig 1G), thereby disrupting contextual expectations. We held constant city layout, object and target counts, the number of agents (56), and exploration time. Only contextual agents' spatial placement varied (congruent vs. incongruent). Thus, normalized GTE differences are not confounded by target availability. This manipulation of the virtual environment and agent placement allowed us to investigate how visual behavior systematically adapts to both environmental layout and agent categories, forming the ground for our entropy analyses of human-context-driven spatial knowledge acquisition.

As a final step, all participants completed a sixth session in which spatial knowledge was assessed. In this session, a VR-based pointing task was administered, where participants indicated the direction of target buildings from multiple randomized locations. Performance was measured as the angular error between the indicated and actual locations. The number and distribution of trials differed between experiments (336 and 224 trials, respectively) to accommodate additional tasks in the second experiment (see [5]). This final task provided a quantitative measure of spatial knowledge, where lower angular errors reflected higher accuracy, enabling direct comparisons across participants and experimental conditions.

## Fixation classification and duration accumulation

The virtual environment was constructed with mesh colliders assigned to all objects, enabling the tracking of participants' fixation behavior by projecting 3D eye vectors onto the scene. Fixations and saccades were identified using a velocity-based classification algorithm [19] adapted for VR environments [20] by incorporating a correction for participants' translational movement [21]. The algorithm segmented continuous data into ten-second intervals, calculating a velocity threshold for each. Fixations were defined as periods of low angular velocity and saccades as periods exceeding calculated velocity thresholds, while accounting for participants' head movements within the VR environment [5]. After classifying all gaze data into fixations and saccades, fixation durations were accumulated per object. For each participant, the total fixation time on each task house and agent was computed by summing the cumulative fixation durations across the five exploration sessions. This yielded a measure of total visual engagement per object, enabling analysis of how attention was focused on specific elements in the virtual environment.

## Entropy calculation

Gaze transition entropy was computed to quantify variability in gaze shifts across different environmental elements. Each participant's transition matrix was generated per session to capture the frequency of fixation transitions between pre-defined categories. Each fixation was assigned to one of eight visual categories: background elements, buildings, task-relevant residential buildings, task-relevant public buildings, global landmarks, acontextual agents, and contextual agents (i.e., congruent and incongruent). Only forward transitions were considered, ensuring that each fixation was linked exclusively to the subsequent fixation in the sequence (see Fig 2A, 2B). The resulting matrices were row-normalized to obtain a probability distribution of fixation transitions across categories (Fig 2C).

To quantify fixation variability, we computed entropy from the row-wise transition probabilities of each category using the Chao-Shen estimator [22], which accounts for sample-size bias and rare transitions. This correction specifically adjusts for singleton transitions—categories observed only once—thus improving entropy estimation under sparse sampling conditions. To ensure methodological rigor and comparability with prior eye-tracking research, we followed the implementation approach used by Wilming et al. [23]. Conceptually, this method aligns with missing-mass approximations in neural modeling frameworks, such as those developed by Haslinger et al. [24], which similarly address the influence of unobserved elements on model structure and entropy-based inference. The category-specific corrected entropy was defined as shown in Eq (1):

$$H(X) = -\sum_x \frac{p_{\text{adj}}(x) \log_2 p_{\text{adj}}(x)}{\text{la}(x)} \qquad (1)$$

Here, $p_{\text{adj}}(x)$ represents the adjusted transition probability for category $x$, and $la(x)$ is the likelihood of observing that category at least once. The adjustment is based on the estimated sample coverage, computed using the number of singleton transitions $S_1$ and the total number of transitions $N$, as shown in Eq (2):

$$C(X) = 1 - \frac{S_1}{N} \qquad (2)$$

To derive a global measure of gaze entropy across all categories, we computed a weighted average of the category-specific entropies. Each weight corresponds to the stationary distribution of gaze within that category, yielding a global GTE score as defined in Eq (3):

$$H_{\text{global}} = \sum \pi(x) H(x) \qquad (3)$$

where $\pi(x)$ denotes the stationary probability of fixating on category $x$, derived from the normalized transition matrix.

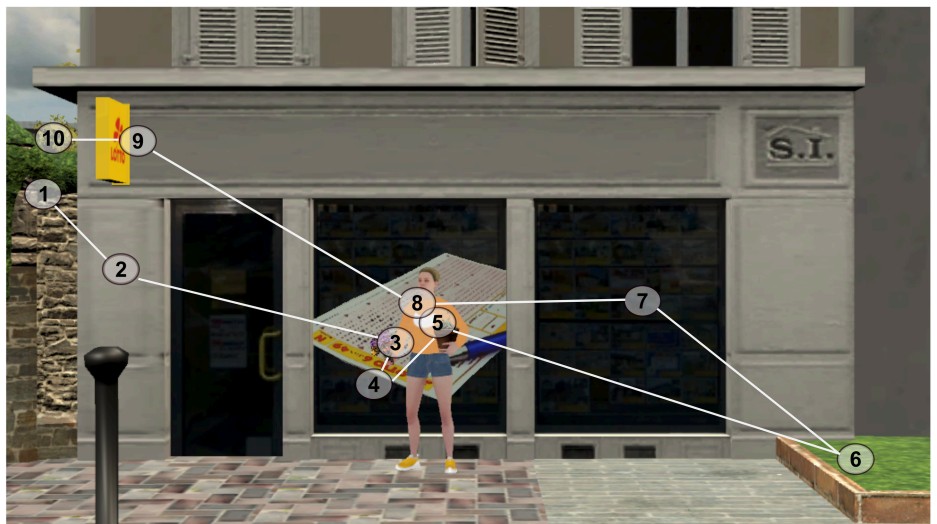

**A**

**B** Transition Counts
**C** Transition Probabilities
**D** GTE Heatmap

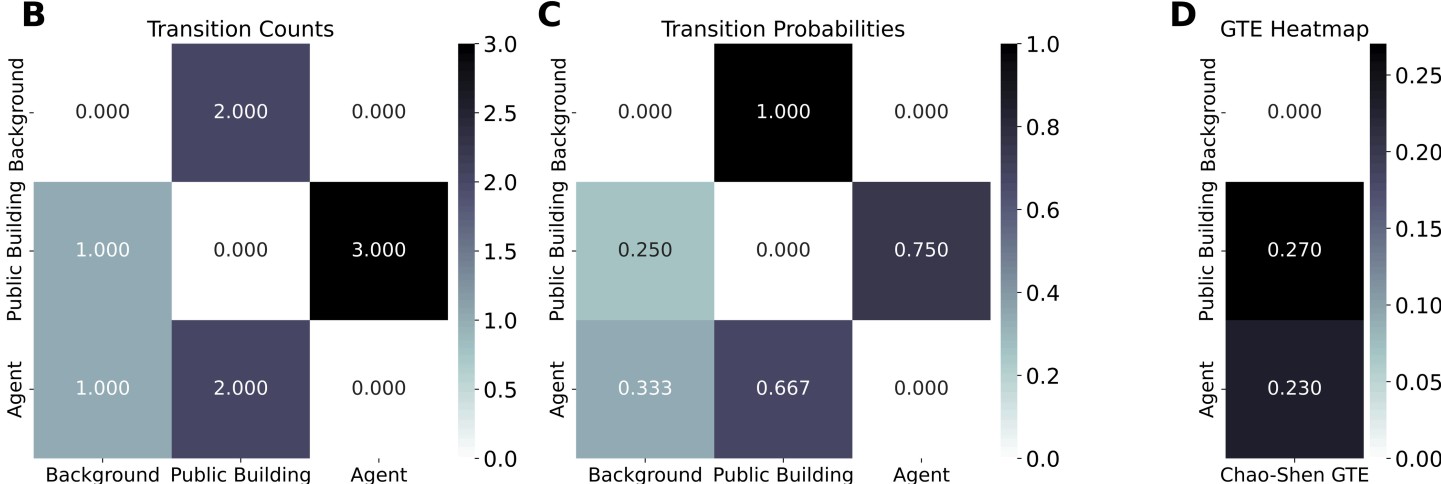

**Fig 2. Illustrative example of fixation transitions across visual categories.** (A) Sequential gaze transitions between visual categories, with numbered circles indicating fixations. (B) Raw transition counts showing gaze shift frequencies. This panel provides a simplified subset for illustrative purposes. Categories not visible in B are excluded from the example. (C) Row-normalized transition probabilities. (D) Chao-Shen corrected gaze transition entropy (GTE), where higher values reflect more variable scanning across categories.

Finally, to allow comparison across participants and conditions, all entropy values were normalized by dividing them by the theoretical maximum entropy given the number of fixation categories $k$. This normalization is shown in Eq (4):

$$H_{\text{norm}} = \frac{H}{\log_2(k)} \qquad (4)$$

Normalized GTE values ranged from 0 (completely predictable transitions) to 1 (maximally unpredictable, uniform transitions), facilitating cross-subject and cross-session analyses.

## Local entropy: Segmented entropy calculation for agent encounters

We introduced a more localized GTE calculation to assess whether gaze variability changed following interactions with human agents. For each detected fixation on a human agent, a 30-second pre-interaction window and a 30-second post-interaction window were defined. Non-overlapping trials were enforced by considering only the first valid fixation on a given collider within 30 seconds. If multiple fixations occurred in rapid succession, only the most recent non-overlapping instance was retained.

Transition matrices were computed separately for each window and normalized to obtain transition probabilities. GTE was then computed using the Chao-Shen correction (see Eq (1)), and values were normalized using Eq (4) to ensure comparability across trials. This method allowed us to systematically assess whether local interactions with agents triggered changes in fixation entropy, indicating shifts in visual exploration strategies.

## Data modeling: Statistical analysis

We used Bayesian hierarchical models to analyze fixation behavior and gaze entropy. All models were implemented in the `brms` package (version 2.21.0) in R [25], which interfaces with Stan for Hamiltonian Monte Carlo (HMC) sampling [26]. Each model was run with four chains of 6,000 iterations, including 3,000 warmup iterations.

Model families were chosen to reflect the distributional properties of each outcome variable. Gamma regression with a log link was used for models predicting absolute pointing error and dwelling time, both of which are strictly positive and right-skewed. Beta regression with a logit link was used for models predicting normalized GTE, given the bounded nature of entropy values between zero and one. For the Gamma models, weakly informative priors were specified explicitly: Normal priors $\mathcal{N}(0, 1)$ were assigned to fixed effects, and Cauchy priors $\text{Cauchy}(0, 2.5)$ were used for intercepts and group-level standard deviations. For the Beta model of GTE, we used weakly informative priors: Normal(0,1) for population-level coefficients (class "b"), Student-$t(3,0,2.5)$ for group-level standard deviations (class "sd"), and Student-$t(3,0,0.8)$ for smooth-term standard deviations (class "sds"). Because Beta likelihoods require strictly interior responses, we mapped $H \in [0, 1]$ to the open unit interval using the Smithson–Verkuilen transform $H_i^* = (H_i(n-1)+0.5)/n$, where $n$ is the analysis-sample size; the same mapping was applied in all Beta models. Posterior distributions were summarized using means and 89% highest density intervals (HDIs), following the default convention in `brms`. Model convergence was assessed using the Gelman–Rubin statistic ($\hat{R} < 1.01$) and visual inspection of trace plots.

**Bayesian analysis of dwelling time on agents and buildings.** To examine how object type ($O_i$: agent vs. building) and agent congruency (treatment dummies $T_i^{(C)}$, $T_i^{(I)}$ with acontextual as reference) influenced dwelling time $D$, we fit a Bayesian hierarchical Gamma model with a log link, including their interaction, and nested random intercepts for participant $p[i]$ and session within participant $p\!:\!s[i]$ (Eq (5)).

$$
\begin{aligned}
D_i &\sim \text{Gamma}\left(\kappa, \frac{\kappa}{\mu_i}\right), \\
\log \mu_i &= \beta_0 + \beta_O O_i + \beta_C T_i^{(C)} + \beta_I T_i^{(I)} \\
&\quad + \beta_{OC}\left(O_i T_i^{(C)}\right) + \beta_{OI}\left(O_i T_i^{(I)}\right) \\
&\quad + u_{p[i]} + v_{p:s[i]}, \\
u_p &\overset{\text{iid}}{\sim} \mathcal{N}(0, \sigma_p^2), \quad v_{p:s} \overset{\text{iid}}{\sim} \mathcal{N}(0, \sigma_{ps}^2), \quad u_p \perp v_{p:s}.
\end{aligned}
\tag{5}
$$

In this model, we employed a log–mean scale with buildings ($O_i$=0) and acontextual as the reference for $T_i^{(C)}$ and $T_i^{(I)}$. Here, $\beta_0$ is the baseline (building × acontextual). $\beta_O$ is the agent–building contrast at acontextual. $\beta_C$ and $\beta_I$ are the congruent–acontextual and incongruent–acontextual contrasts among buildings. $\beta_{OC}$ and $\beta_{OI}$ are the additional congruent/incongruent effects for agents beyond those for buildings. Effects are reported as multiplicative factors $\exp(\delta)$

(percent change = $100[\exp(\delta) - 1]\%$). Baseline heterogeneity was modeled via random intercepts $u_{p[i]} \sim \mathcal{N}(0, \sigma_p^2)$ and $v_{p:s[i]} \sim \mathcal{N}(0, \sigma_{ps}^2)$, assumed independent ($u_p \perp v_{p:s}$), where $p[i]$ indexes participants and $p:s[i]$ indexes sessions nested within participants.

**Bayesian time models for agent-locked entropy dynamics.** To model the evolution of fixation entropy $H$ across agent encounters, we implemented a beta regression with a logit link. The model incorporated encounter index $k$ and a smoothing spline $s(k,5)$ to account for nonlinear temporal changes, as shown in Eq (6). We used treatment coding for the pre/post factor, $P_i \in \{0, 1\}$ with pre = 0 and post = 1, to keep the intercept interpretable as the pre baseline and the main coefficient as the post–pre change.

$$
\begin{aligned}
H_i^* &\sim \text{Beta}(\mu_i \phi, (1 - \mu_i)\phi), \qquad \phi > 0, \\
\text{logit}(\mu_i) &= \beta_0 + \beta_P P_i + s(k_i; 5) + u_{p[i]} + v_{p:s[i]}, \\
u_p &\overset{\text{iid}}{\sim} \mathcal{N}(0, \sigma_u^2), \qquad v_{p:s} \overset{\text{iid}}{\sim} \mathcal{N}(0, \sigma_v^2), \qquad u_p \perp v_{p:s}.
\end{aligned}
\tag{6}
$$

In this model, $P_i$ contrasts pre- and post-encounter entropy values (pre = 0, post = 1), while the spline $s(k,5)$ captures smooth transitions across successive interactions. Here, $u_{p[i]}$ and $v_{p:s[i]}$ are participant- and session-within-participant–level random intercepts, respectively. The notation $u_p \overset{\text{iid}}{\sim} \mathcal{N}(0, \sigma_u^2)$ and $v_{p:s} \overset{\text{iid}}{\sim} \mathcal{N}(0, \sigma_v^2)$ means that the participant effects $\{u_p\}$ are independent and identically distributed Normal with variance $\sigma_u^2$, and the session effects $\{v_{p:s}\}$ are likewise iid Normal with variance $\sigma_v^2$. The independence statement $u_p \perp v_{p:s}$ specifies that participant- and session-level deviations are mutually independent. Index $p[i]$ maps observation $i$ to its participant $p$, while $p:s[i]$ maps $i$ to the session $s$ nested within $p$. In `brms` terms, this corresponds to `(1 | ID) + (1 | ID:Session)`, i.e., intercept-only varying effects at both levels. Coefficients are estimated on the log-odds scale and are reported as odds ratios $\exp(\delta)$; we also present posterior predictions on the original $(0,1)$ scale for interpretability.

**Post-encounter entropy model.** To isolate the effects of agent context on post-interaction gaze variability, we fit a Beta(logit) model to post-fixation GTE with nested random intercepts for participant and session-within-participant (Eq (7)).

$$
\begin{aligned}
H_i^* &\sim \text{Beta}(\mu_i \phi, (1 - \mu_i)\phi), \\
\text{logit}(\mu_i) &= \beta_0 + \gamma_1 Z_{1i} + \gamma_2 Z_{2i} + u_{p[i]} + v_{p:s[i]}, \\
u_p &\overset{\text{iid}}{\sim} \mathcal{N}(0, \sigma_u^2), \qquad v_{p:s} \overset{\text{iid}}{\sim} \mathcal{N}(0, \sigma_v^2), \qquad u_p \perp v_{p:s}.
\end{aligned}
\tag{7}
$$

Here, $H_i^* \in (0, 1)$ is the Smithson–Verkuilen–adjusted entropy, $\phi > 0$ is the Beta precision, and the linear predictor is on the logit scale. Because we treat the agent encounter as the common driver of the entropy change, we encoded context with sum-to-zero contrasts. This makes the intercept the grand mean across acontextual, congruent, and incongruent encounters, and each coefficient a context-specific shift around that mean. With sum coding and level order {acontextual, congruent, incongruent}, the two contrast columns are $(Z_1, Z_2) = (1, 0)$ for acontextual, $(0, 1)$ for congruent, and $(-1, -1)$ for incongruent. Thus the intercept $\beta_0$ is the grand mean (logit scale), and the level-specific logit means are

$$
\eta_A = \beta_0 + \gamma_1, \quad \eta_C = \beta_0 + \gamma_2, \quad \eta_I = \beta_0 - \gamma_1 - \gamma_2.
$$

Pairwise log-odds contrasts follow as linear combinations, e.g. $\eta_C - \eta_A = \gamma_2 - \gamma_1$, $\eta_I - \eta_A = -2\gamma_1 - \gamma_2$, and $\eta_C - \eta_I = \gamma_1 + 2\gamma_2$. We report effects on the log-odds scale, as odds ratios $\exp(\Delta)$, and as posterior predictions on the probability scale via the inverse logit.

**Entropy as a predictor of performance.** To explain variation in spatial recall performance, we modeled trial-wise absolute pointing error ($Y_i$) as a function of agent type, building type, and post-encounter gaze dynamics. On each trial

of the pointing-to-building task, participants indicated the direction to a target building from one of 28 predefined pointing locations.

$$Y_i \sim \mathrm{Gamma}\left(\kappa_Y, \frac{\kappa_Y}{\mu_{Yi}}\right),$$

$$\log \mu_{Yi} = \beta_0^{(Y)} + \beta_C^{(Y)} C_i + \beta_I^{(Y)} I_i + \beta_{\mathrm{pub}}^{(Y)} P_i$$
$$+ \beta_{\mathrm{GTE}}^{(Y)} M_i + \beta_{\mathrm{DW-A}}^{(Y)} \mathrm{DW}_{i,\mathrm{agent}}^{(z)} + \beta_{\mathrm{DW-B}}^{(Y)} \mathrm{DW}_{i,\mathrm{building}}^{(z)} + u_{p[i]}^{(Y)} + v_{s[i]}^{(Y)},$$

$$u_p^{(Y)} \overset{\mathrm{iid}}{\sim} \mathcal{N}(0, \sigma_{u,Y}^2), \qquad v_s^{(Y)} \overset{\mathrm{iid}}{\sim} \mathcal{N}(0, \sigma_{v,Y}^2), \qquad u_p^{(Y)} \perp v_s^{(Y)}.$$

(8)

Here $Y_i$ denotes absolute pointing error for observation $i$. The outcome followed a Gamma likelihood with a log link, and the linear predictor included: (i) agent type in front of the target, encoded as treatment dummies $C_i = \mathbb{I}[\mathrm{congruent}]$ and $I_i = \mathbb{I}[\mathrm{incongruent}]$ (acontextual reference); (ii) building type, $P_i = \mathbb{I}[\mathrm{public}]$ (residential reference); (iii) post-encounter gaze transition entropy $M_i \equiv \mathrm{GTE}_{i,\mathrm{post}}$ on the $[0,1]$ scale used in the mediator model; and (iv) standardized dwelling time $\mathrm{DW}_{i,\mathrm{agent}}^{(z)}$ and $\mathrm{DW}_{i,\mathrm{building}}^{(z)}$ (total fixation durations, $z$-scored). Crossed random intercepts captured heterogeneity for participant $p[i]$ and starting location $s[i]$. Under the log link, $\exp(\beta)$ yields the multiplicative change in expected error.

**Counterfactual mediation with a single mediator (GTE).** We use causal mediation to ask: to what extent do agent effects on pointing performance operate via changes in post-encounter gaze transition entropy (GTE), and to what extent would they persist if GTE were held at its acontextual level? In the explanatory framework, these correspond to the natural indirect effect (NIE), the natural direct effect (NDE), and their sum (TE), defined in Eq (14).

Agent type $A$ is coded with baseline $a_0$=acontextual and treatment levels $a_1 \in \{\mathrm{congruent}, \mathrm{incongruent}\}$. Estimation follows Bayesian parametric $g$-computation: we fit a Beta–logit mediator model for normalized post-encounter GTE (Eq (9)) and a Gamma–log outcome model for expected absolute error (Eq (8)). For each replication, we independently pair one mediator-model draw with one outcome-model draw, compute the mediator mean under each $(a_k, c)$ via Eq (11a), and plug it into the outcome predictor and mean (Eqs (11b)—(11c)). Dwelling time covariates are fixed at their standardized means by setting the $z$-scored terms to 0, and predictions are population-level by setting random intercepts to zero (average participant and starting location). We then average over building type using empirical weights (Eq (12); cf. Eq (10)) to form $Y_{jk,d}$, map these to $Y_{00}, Y_{10}, Y_{11}$ (Eq (13)), and compute NDE, NIE, and TE (and their mean-ratio counterparts) per Eq (14). This implements the parametric $g$-formula [27–29] within a `brms`/Stan workflow [25,26].

In the mediator model, coefficients are on the log-odds scale (reported as odds ratios via $\exp(\beta)$), and no $A{\times}M$ interaction is included, so agent effects on the mediator are additive on the logit scale. Because the outcome uses a log link, the reported ratios in Eq (14) quantify proportional changes in the expected error on the mean scale.

**Mediator model (GTE).** We consider agent type with baseline $a_0$=acontextual and a generic treatment level $a_1 \in \{\mathrm{congruent}, \mathrm{incongruent}\}$. All counterfactual quantities below are computed separately for $a_1$=congruent and for $a_1$=incongruent; we report results for both contrasts. Let $M_i$ denote normalized GTE for observation $i$ and $M_i^*$ its Smithson–Verkuilen mapping to $(0,1)$. For compact notation in displayed equations, define indicator dummies $C_i=\mathbb{I}[A_i=\mathrm{congruent}]$, $I_i=\mathbb{I}[A_i=\mathrm{incongruent}]$, and $P_i=\mathbb{I}[\mathrm{Context}_i=\mathrm{public}]$. We model $M_i^*$ with a Beta likelihood and logit link using these contrasts, with crossed random intercepts for participant $p[i]$ and starting location $s[i]$. Parameters with superscript $(Y)$ below belong to the outcome model; parameters without a superscript belong to the mediator model.

$$M_i^* \sim \mathrm{Beta}\big(\mu_i \phi, (1 - \mu_i)\phi\big), \qquad \phi > 0,$$

$$\mathrm{logit}(\mu_i) = \beta_0 + \beta_C C_i + \beta_I I_i + \beta_{\mathrm{pub}} P_i + u_{p[i]}^{(M)} + v_{s[i]}^{(M)},$$

$$u_p^{(M)} \overset{\mathrm{iid}}{\sim} \mathcal{N}(0, \sigma_{u,M}^2), \qquad v_s^{(M)} \overset{\mathrm{iid}}{\sim} \mathcal{N}(0, \sigma_{v,M}^2), \qquad u_p^{(M)} \perp v_s^{(M)}.$$

(9)

 

Here, $\beta_0$ is the population-level log-odds mean for acontextual agents in residential context (for average participant and location, $u=v=0$). The precision parameter $\phi$ is estimated and governs dispersion around $\mu_i$. Coefficients are on the logit scale and are reported as odds ratios via $\exp(\beta)$.

**Counterfactual setup and context averaging.** Dwelling time covariates (agent dwelling, building dwelling) in the outcome model were z-scored on the analysis dataset, so that 0 corresponds to the sample mean (SD = 1). Because performance also depends on building type, predictions are averaged over residential and public using empirical weights $w_{\text{res}}, w_{\text{pub}} \geq 0$ with $w_{\text{res}} + w_{\text{pub}} = 1$. For the agent level $j \in \{0, 1\}$ used in the outcome model ($j=0 \Rightarrow a_0$, $j=1 \Rightarrow a_1$) and the agent level $k \in \{0, 1\}$ used to generate the mediator ($k=0 \Rightarrow a_0$, $k=1 \Rightarrow a_1$), we define

$$Y_{jk} = w_{\text{res}}\, Y_{jk}^{(\text{residential})} + w_{\text{pub}}\, Y_{jk}^{(\text{public})}. \tag{10}$$

Here, $Y_{jk}^{(\text{residential})}$ and $Y_{jk}^{(\text{public})}$ are expected errors under residential and public, respectively, with the mediator generated as if $A=a_k$ and the outcome evaluated as if $A=a_j$. In practice, $w_{\text{res}}$ and $w_{\text{pub}}$ equal the observed proportions of residential and public trials.

**Per-draw estimation.** For each posterior draw $d$, we first compute the mediator mean under agent level $a_k$ and context $c \in \{\text{residential}, \text{public}\}$ from Eq (9), and then plug that mean into the performance model (Eq (8)) evaluated at agent level $a_j$. dwelling time covariates are fixed at their standardized means by setting them to 0 on the z-scale, and predictions are made at the population level by setting random intercepts to zero (average participant and starting location). For compactness, let $C_t=\mathbb{1}[a_t=\text{congruent}]$, $I_t=\mathbb{1}[a_t=\text{incongruent}]$ for $t \in \{j, k\}$, and $P_c=\mathbb{1}[c=\text{public}]$. This yields

$$\mu_{k,c,d}^{(M)} = \text{logit}^{-1}\left(\beta_{0,d} + \beta_{C,d}\, C_k + \beta_{I,d}\, I_k + \beta_{\text{pub},d}\, P_c\right), \tag{11a}$$

$$\eta_{j,k,c,d} = \beta_{0,d}^{(Y)} + \beta_{C,d}^{(Y)}\, C_j + \beta_{I,d}^{(Y)}\, I_j + \beta_{\text{pub},d}^{(Y)}\, P_c + \beta_{\text{GTE},d}^{(Y)}\, \mu_{k,c,d}^{(M)}, \tag{11b}$$

$$\mu_{j,k,c,d}^{(Y)} = \exp(\eta_{j,k,c,d}). \tag{11c}$$

In Eq (11a), $\mu_{k,c,d}^{(M)}$ is the mediator mean on the probability scale for agent $a_k$ and context $c$ in draw $d$; $\text{logit}^{-1}(\cdot)$ maps the linear predictor to $(0,1)$. The indicators $C_k$, $I_k$, and $P_c$ select agent and context contrasts; $\beta_{0,d}, \beta_{C,d}, \beta_{I,d}, \beta_{\text{pub},d}$ are the corresponding fixed effects from the mediator model in draw $d$. The Beta precision $\phi$ affects dispersion but not the plug-in mean, so stochastic Beta draws are unnecessary for point predictions. In Eq (11b), the outcome linear predictor combines the agent and context contrasts with the mediator mean via $\beta_{\text{GTE},d}^{(Y)}$, and Eq (11c) returns the expected error on the original scale under the Gamma–log mean.

Uncertainty is propagated by Bayesian parametric g-computation: for each replication, we independently sample one mediator-model draw and one outcome-model draw (random pairing), compute the plug-in mediator mean $\mu_{k,c,d}^{(M)}$ and the corresponding outcome mean $\mu_{j,k,c,d}^{(Y)}$ using the same context indicator $c$ in both steps, and then average over contexts as specified in Eq (10).

In Eq (11b), $\eta_{j,k,c,d}$ is the outcome-model linear predictor on the log scale when the outcome exposure is set to $a_j$ and the mediator is set to its mean under $a_k$ and context $c$ from Eq (11a). The terms $\beta_{C,d}^{(Y)}, \beta_{I,d}^{(Y)}, \beta_{\text{pub},d}^{(Y)}$ are the draw-specific fixed effects for the outcome model's agent and context contrasts; $\beta_{\text{GTE},d}^{(Y)}$ is the outcome-model slope linking GTE to expected pointing error. Using the same $c$ in both the mediator and outcome pieces maintains coherent conditioning on context. dwelling time covariates are fixed to zero, and random intercepts are set to zero to obtain predictions for an average participant and starting location.

Finally, Eq (11c) exponentiates the linear predictor to return $\mu^{(Y)}_{j,k,c,d}$, the expected absolute pointing error on the original outcome scale under the Gamma–log model (i.e., $\mathbb{E}[Y \mid a_j, M=\mu^{(M)}_{k,c,d}, c]$). These quantities are then averaged over contexts via Eq (10) to form $Y_{jk,d}$ for use in the counterfactual contrasts.

For brevity in subscripts, res and pub denote residential and public. Context-marginal potential outcomes were then formed via Eq (10):

$$Y_{jk,d} = w_{\text{res}}\,\mu^{(Y)}_{j,k,\text{res},d} \;+\; w_{\text{pub}}\,\mu^{(Y)}_{j,k,\text{pub},d}. \tag{12}$$

Eq (12) defines the context–marginal potential outcome for a given exposure/mediator combination $(j,k)$ in posterior draw $d$. The terms $\mu^{(Y)}_{j,k,\text{res},d}$ and $\mu^{(Y)}_{j,k,\text{pub},d}$ are the expected pointing errors from the performance model (Eq (8)) evaluated under the residential and public contexts, respectively, when the outcome is set as if $A=a_j$ and the mediator is generated as if $A=a_k$. The weights $w_{\text{res}}$ and $w_{\text{pub}}$ are the empirical frequencies of residential and public trials (nonnegative, summing to 1).

**Counterfactual outcomes and effects.** To simplify notation, let the context-averaged potential outcomes be

$$
\begin{aligned}
Y_{00} &\equiv Y_{j=0,k=0}, \\
Y_{10} &\equiv Y_{j=1,k=0}, \\
Y_{11} &\equiv Y_{j=1,k=1}.
\end{aligned}
\tag{13}
$$

Eq (13) maps these context–marginal outcomes to the three counterfactuals used in mediation analysis. $Y_{00}$ is the expected error when both the outcome model and the mediator are evaluated as if the agent were acontextual $(j=0, k=0)$. $Y_{10}$ holds the mediator at its acontextual distribution but evaluates the outcome model as if the agent were at $a_1$ $(j=1, k=0)$; this isolates the part of the effect not operating through changes in the mediator. $Y_{11}$ evaluates both mediator and outcome as if the agent were at $a_1$ $(j=1, k=1)$, capturing the fully treated scenario.

Natural direct, indirect, and total effects, together with multiplicative summaries (risk ratios), are

$$\text{NDE} = Y_{10} - Y_{00}, \qquad\qquad RR_{\text{NDE}} = \frac{Y_{10}}{Y_{00}}, \tag{14a}$$

$$\text{NIE} = Y_{11} - Y_{10}, \qquad\qquad RR_{\text{NIE}} = \frac{Y_{11}}{Y_{10}}, \tag{14b}$$

$$\text{TE} = Y_{11} - Y_{00}, \qquad\qquad RR_{\text{TE}} = \frac{Y_{11}}{Y_{00}}. \tag{14c}$$

Eq (14) reports the effect decompositions. Because the outcome model uses a log link, the risk ratios quantify proportional changes in expected error. For clarity, the draw index $d$ is suppressed in Eq (14); in computation, effects are formed per draw and summarized by posterior means with 95% credible intervals.

## Results

To examine how contextual agents influenced gaze behavior and spatial knowledge, we analyzed fixation durations, gaze transition entropy (GTE), and pointing accuracy using Bayesian hierarchical models. These analyses were conducted on gaze data recorded across five exploration sessions and spatial knowledge assessment from a final test session. Fixation metrics were computed at both global and local levels, with separate models addressing cumulative dwelling time, entropy dynamics over time, and post-encounter entropy changes. All models included participant- and session-level random effects to account for individual variability. For performance analyses, we used crossed random intercepts for participant and pointing location. Performance (absolute pointing error) was modeled with a Gamma likelihood and log link, including

post-encounter GTE, $z$-scored dwelling time on agents and buildings, agent type (acontextual, congruent, incongruent), and building type (public vs. residential) as predictors. We then estimated a single-mediator (GTE) counterfactual mediation using Bayesian parametric $g$-computation, holding dwelling time at its mean and averaging over building type. Below, we present the results in five parts: (1) fixation duration and engagement across agent conditions, (2) temporal modeling of entropy dynamics, (3) post-interaction changes in visual exploration, (4) performance prediction from GTE, dwelling times, agent type, and building type, and (5) a single-mediator counterfactual mediation quantifying the GTE pathway.

**Fixation duration and engagement across agent conditions.** To examine whether agent congruency influenced attentional engagement, we analyzed total dwelling time and fixation counts on agents and associated buildings across conditions (Fig 3). We hypothesized that incongruent agents would elicit the greatest attention, followed by congruent and

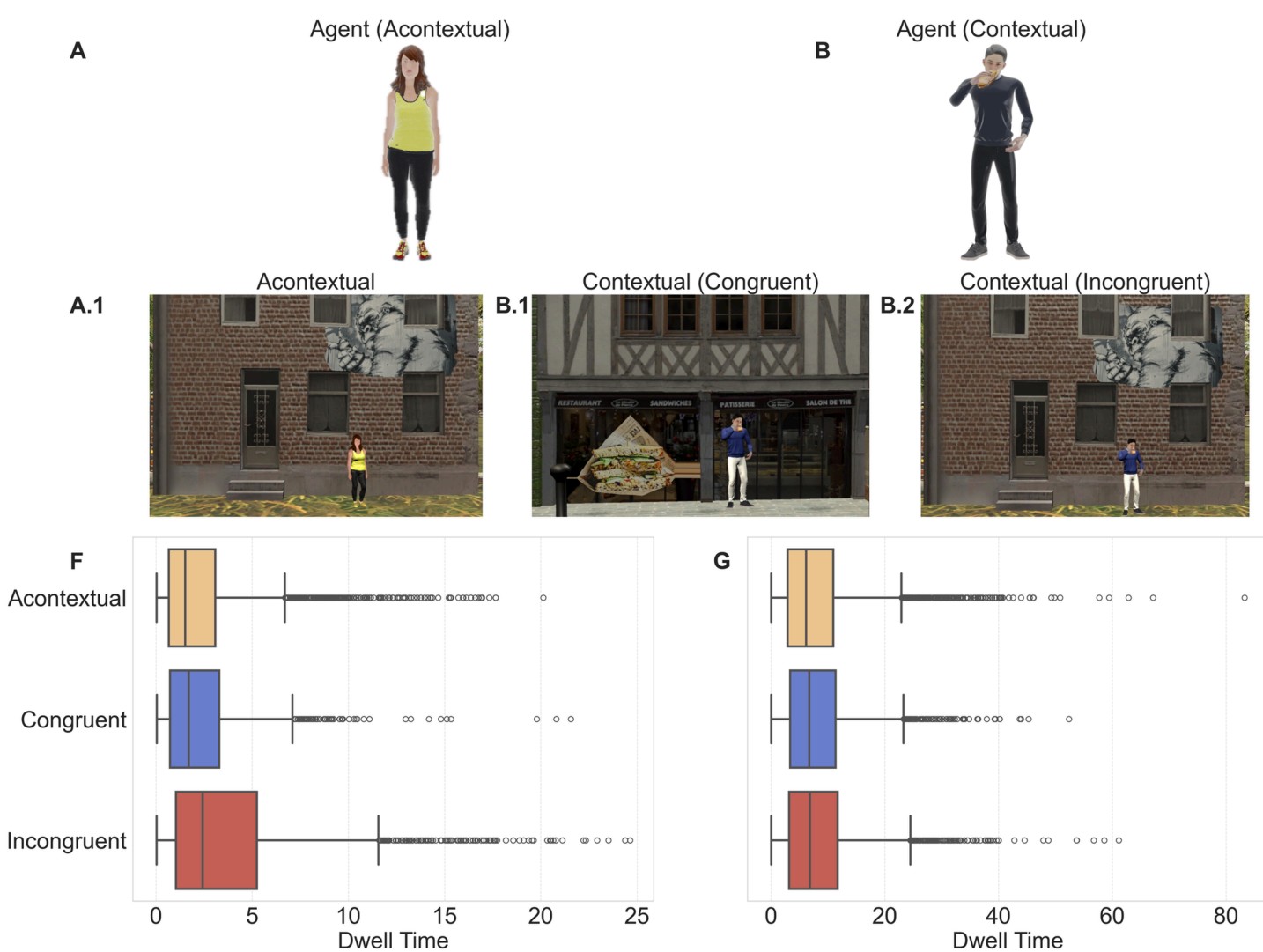

**Fig 3. Agent types and dwelling time distributions.** (A–B) Agent type definitions. (A) Acontextual agent: avatar standing neutrally without any object. (B) Contextual agent: avatar interacting with an object; here, eating a sandwich. (A.1, B.1, B.2) Scene examples for the three conditions: (A.1) Acontextual, (B.1) Congruent, (B.2) Incongruent. In (B.1), the agent's object matches the environment; in (B.2), it does not. (F–G) Boxplots of dwelling time for (F) agent fixations and (G) building fixations.

acontextual agents, and that these effects might extend to nearby buildings. Unless otherwise noted, descriptive summary statistics (M, SD) are calculated across participants within each agent condition. For each participant, dwelling time and fixation counts were aggregated within condition and then summarized across participants. Visual engagement was assessed through descriptive statistics and a Bayesian hierarchical regression model.

Dwelling time is a common proxy for attentional allocation, with longer durations reflecting greater scrutiny or interest [30]. Incongruent agents attracted longer fixations (M = 3.83 s, SD = 3.90) than both acontextual (M = 2.36 s, SD = 2.57) and congruent agents (M = 2.26 s, SD = 2.50) (Fig 3). Similarly, fixation counts were highest for incongruent agents (M = 7.90, SD = 10.52), compared to acontextual (M = 5.19, SD = 6.75) and congruent agents (M = 5.09, SD = 7.53). These patterns suggest increased visual engagement with incongruent agents, likely due to their violation of contextual expectations.

We next assessed whether agent congruency influenced attention toward nearby buildings. Participants spent a comparable amount of time fixating on buildings near congruent (M = 8.20 s, SD = 6.54) and incongruent agents (M = 8.50 s, SD = 7.40), while slightly less time was spent near acontextual agents (M = 8.00 s, SD = 7.13) (Fig 3). Fixation counts followed a similar trend: 14.90 (SD = 14.18) for congruent, 15.22 (SD = 15.54) for incongruent, and 14.75 (SD = 14.81) for acontextual conditions. These small differences suggest that while incongruent agents attracted greater direct attention, spillover effects on nearby buildings were less pronounced.

To formally assess the effect of agent congruency and object type on visual engagement, we fit a Bayesian hierarchical Gamma model with a log link including their interaction and varying intercepts for participants and participant-by-session. The Gamma shape parameter $\kappa$ indicated the expected right-skew in dwelling times (shape = 1.49, 95% CrI [1.46, 1.52]).

Relative to buildings in the acontextual condition (reference), agents were associated with shorter dwelling time overall ($\beta_{Agent} = -1.23$, 95% CrI [−1.33, −1.26]), reflecting a 72.7% reduction compared to buildings. Incongruency had a small but positive effect on overall dwelling time ($\beta = 0.08$, CrI: [0.04, 0.12]), corresponding to an 8.3% increase Importantly, this effect was moderated by object type (agent×congruent $\beta = 0.06$, 95% CrI [−0.00, 0.12]; agent×incongruent $\beta = 0.41$, 95% CrI [0.36, 0.47]), such that incongruency increased dwelling time for agents while leaving buildings largely unchanged. Specifically, incongruent agents received 52% longer fixations (congruent 15%), whereas nearby buildings changed by 1% (uncertain) in incongruent scenes and 8% in congruent scenes. These results reveal a directional asymmetry in attention allocation: participants engaged more strongly with socially incongruent agents while maintaining comparable dwelling time on the surroundings. This pattern suggests that violations of social expectations capture attention more effectively than mismatches between an object and its environment.

These results reveal a directional asymmetry in attention allocation. Participants engaged more strongly with socially incongruent agents (+53% dwelling time vs. acontextual agents; 89% HDI: +42%–+65%), and also showed a credible but smaller increase for congruent agents (+14.5%; 89% HDI: +5.6%–+24.1%). By contrast, for nearby buildings, congruent scenes elicited longer dwelling time than acontextual (+8.3%; 89% HDI: +4.9%–+12.0%), while incongruent scenes were essentially unchanged relative to acontextual (+1.2%; 89% HDI spans zero) and therefore 7% lower than congruent. This pattern is consistent with expectancy violations, concentrating attention on the agent rather than the background building.

**Gaze transition entropy across visual categories and congruency.** To assess whether agent congruency influenced the fluidity of visual exploration, we examined GTE across visual categories using Chao-Shen corrected entropy estimates (Fig 4). We hypothesized that incongruent agents would disrupt attentional flow, leading to greater variability in gaze transitions.

On average, overall entropy was higher in the incongruent condition ($M = 0.44$, $SD = 0.05$) than in the congruent condition ($M = 0.42$, $SD = 0.05$). This pattern was most pronounced when focusing on agents. GTE for agents rose from $M = 0.63$, $SD = 0.09$ in the congruent condition to $M = 0.67$, $SD = 0.11$ in the incongruent condition, while the acontextual agents approximated the congruent condition ($M = 0.63$, $SD = 0.09$). These values reflect more exploratory and less predictable transitions following encounters with incongruent agents relative to both acontextual and congruent contexts.

PLOS Computational Biology

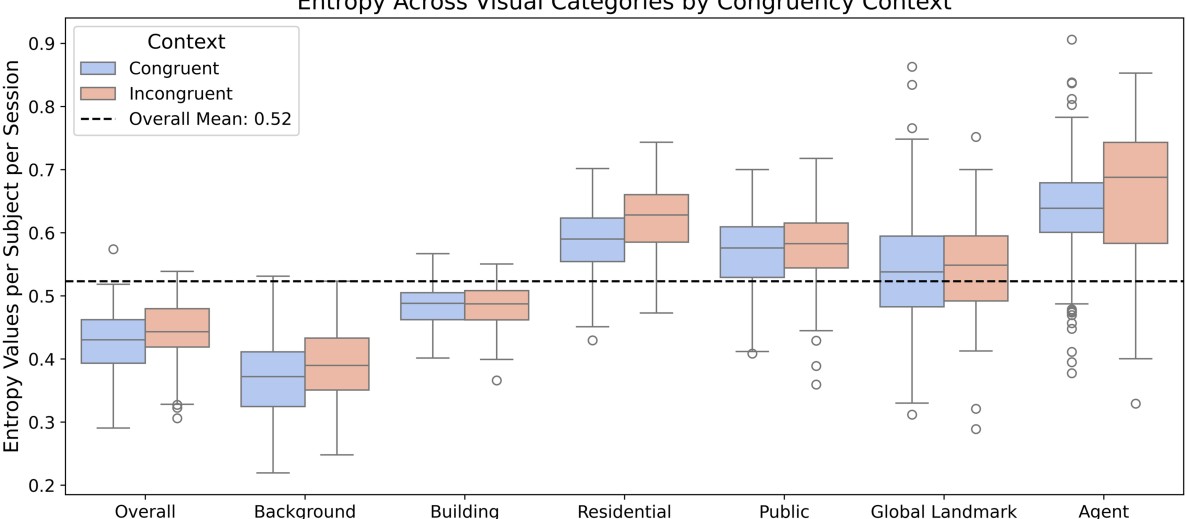

**Fig 4**. **Categorical gaze transition entropy across visual categories and contexts.** (A) Boxplots of Chao-Shen corrected gaze transition entropy (GTE) per subject per session, split by visual category and condition. Higher values indicate more unpredictable gaze transitions. The dashed line indicates the overall mean.

In contrast, entropy estimates for other categories were largely stable across conditions. Buildings showed no meaningful change ($M = 0.48$, $SD = 0.03$ congruent; $M = 0.48$, $SD = 0.04$ incongruent), and global landmarks also remained constant ($M = 0.54$, $SD = 0.10$ vs. $M = 0.54$, $SD = 0.07$).

These findings indicate that incongruent agents uniquely disrupt gaze dynamics, increasing transition entropy and prompting more variable patterns of visual exploration. Other visual elements, including buildings and landmarks, were comparatively unaffected by agent congruency, suggesting a targeted influence of social-contextual anomalies on attentional behavior.

**Exploration–exploitation trade-off: The effect of agent congruency.** To investigate how agent congruency modulated attentional trade-offs between exploration and exploitation, we analyzed the relationship between dwelling time and gaze transition entropy using kernel density estimation (KDE) and correlation analysis (Fig 5). We expected incongruent agents to elicit prolonged and less predictable gaze behavior—indicative of greater exploration.

In Fig 5A, KDE analysis showed that incongruent agents had the highest mode of dwelling time (1.17 s) and centroid (4.19 s), suggesting both common and average fixations were longer compared to congruent (mode = 0.80 s, centroid = 3.21 s) and acontextual agents (mode = 0.76 s, centroid = 2.96 s). Additionally, incongruent agents showed the widest dwelling time distribution (FWHM = 3.82 s), reflecting greater variability than congruent (2.69 s) and acontextual agents (2.31 s).

Entropy patterns mirrored this trend: incongruent agents showed the highest mode (0.735) and centroid (0.661), indicating more dispersed gaze behavior. Congruent agents exhibited the lowest entropy (mode = 0.637, centroid = 0.626) with the narrowest spread (FWHM = 0.094), suggesting fixations on them were more structured and predictable. These results suggest that incongruence increased both the duration and variability of gaze, reflecting a shift toward more exploratory viewing strategies.

To assess the relationship between visual exploitation and exploration, we computed correlation matrices between dwelling time, fixation count, and GTE (Fig 5B– 5D). For acontextual agents, fixation count and dwelling time were moderately correlated ($r = 0.50$), while both were negatively associated with GTE ($r = -0.61$; $r = -0.19$), suggesting that greater local attention tended to co-occur with more structured gaze patterns.

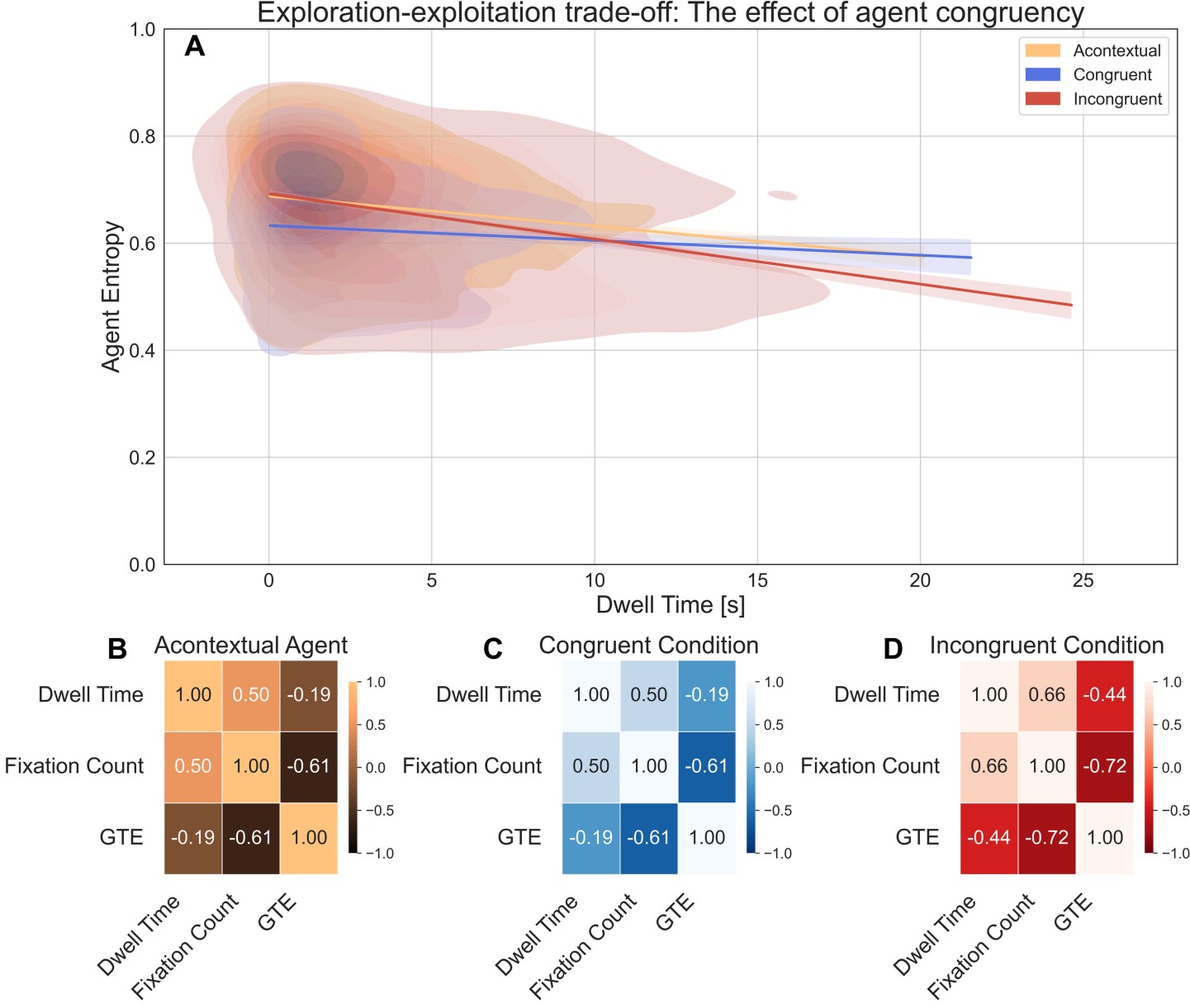

**Fig 5**. **Relationship between dwelling time, gaze transition entropy, and fixation count for agent categories.** (A) Kernel density estimation (KDE) plot of dwelling time vs. GTE for (yellow) acontextual, (blue) congruent, and (red) incongruent agents. Regression lines show trends per condition. (B–D) Correlation matrices of dwelling time, fixation count, and GTE.

In the congruent condition, fixation count and dwelling time were similarly correlated ($r = 0.50$), and both showed weaker negative correlations with GTE ($r = -0.61$; $r = -0.19$), indicating a similar but less pronounced trade-off between attention and entropy. In the incongruent condition, these trade-offs were most pronounced. Fixation count and dwelling time were strongly correlated ($r = 0.66$), while their correlations with GTE were sharply negative ($r = -0.72$; $r = -0.44$). These findings suggest that incongruent agents elicited a robust attentional capture effect, with prolonged fixations tightly coupled to reduced gaze variability, indicating a sharper shift toward exploitation. Together, these results highlight how

agent congruency modulates attentional dynamics: incongruence enhances dwelling time while simultaneously suppressing exploratory gaze, producing the strongest exploration–exploitation trade-off across conditions.

**Time analysis of GTE: Agent effects on entropy.** To assess whether agent encounters dynamically influenced visual exploration, we analyzed gaze transition entropy (GTE) before and after fixating on an agent. We fit a Bayesian hierarchical regression with a smoothing spline for event index and a multilevel structure for individual and session variation to examine entropy trajectories over time.

As shown in Fig 6A, post-fixation entropy increased on the logit scale ($\beta = 0.38$, 95% CI = [0.36, 0.40]). Back-transforming using the model's intercept ($\alpha = 0.55$) suggests entropy rose from approximately 0.63 (pre-fixation) to 0.72 (post-fixation), a relative increase of 13.1%. This indicates that gaze transitions became more dispersed and less predictable after fixating on agents. The smoothing spline for event index showed a trivial effect ($\beta = 0.20$, 95% CI = [-0.14, 0.58]), suggesting no systematic drift over successive encounters.

To ensure effects reflected genuine changes rather than temporal drift, pre- and post-fixation windows were defined as non-overlapping 30-second segments. Modest variability in participant- and session-level intercepts ($SD_{ID} = 0.19$; $SD_{Session} = 0.19$) confirmed baseline stability. A Bayes factor of $BF_{10} \approx 6.30 \times 10^{299}$ strongly favored the full model, confirming a real increase in entropy after agent encounters.

To isolate the influence of agent type, we fit a second Bayesian beta regression model to post-fixation GTE, using effect (sum-to-zero) coding for agent type (Fig 6B). The model intercept (grand mean across agent types) was $\beta = 0.94$

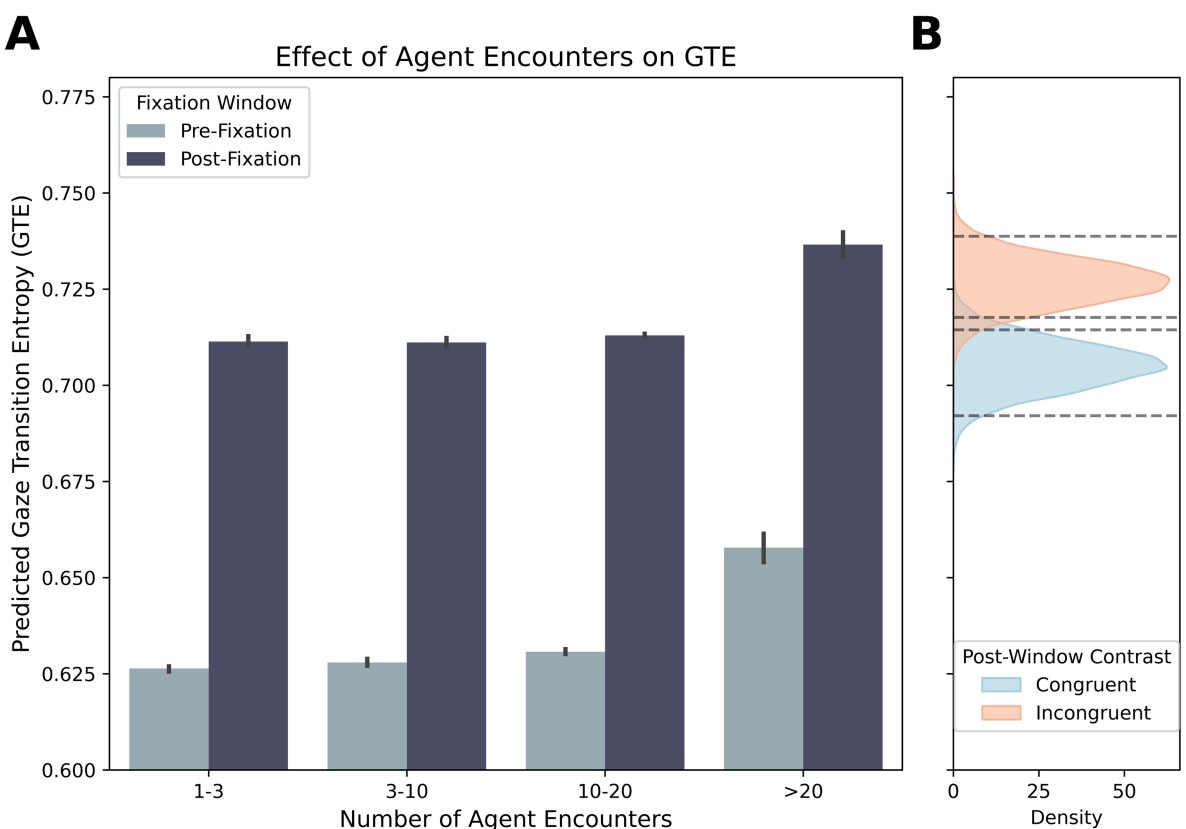

**Fig 6. Effect of agent encounters on gaze transition entropy.** (A) Model-based predictions of GTE before and after agent fixations (shaded ribbons: uncertainty). (B) Posterior distributions from the post-fixation model comparing agent types (dashed lines: 95% intervals).

(95% CI = [0.89, 1.00]), corresponding to a mean GTE of $\sim 0.72$. Compared to this baseline, congruent agents decreased post-fixation entropy ($\Delta = -0.015$; 95% HPD [$-0.021$, $-0.009$]), yielding a back-transformed mean of 0.704. In contrast, incongruent agents increased post-fixation entropy ($\Delta = +0.006$; 95% HPD [0.0001, 0.011]), with a mean of 0.725. These results show that congruent agents lead to more focused gaze patterns, while incongruent agents trigger more exploratory viewing behavior. Together, these findings reveal that agent congruency modulates post-fixation gaze behavior, with incongruent cues promoting exploration and congruent cues reinforcing structured, goal-directed scanning.

**Performance prediction using entropy and dwelling times.** To evaluate how gaze dynamics contribute to spatial learning, we implemented a Bayesian hierarchical Gamma regression model to predict absolute pointing error based on post-encounter gaze entropy, dwelling time (z-scored), agent type, and building type. Random intercepts were included for participants and pointing task starting locations to account for individual and environmental variability. Participant-level variance exceeded that of location-level effects ($SD_{\text{Subject}} = 0.33$, 95% $CrI = [0.28, 0.40]$; $SD_{\text{Location}} = 0.19$, 95% $CrI = [0.14, 0.26]$), indicating that individual differences played a greater role in performance outcomes.

Turning to the fixed effects (Fig. 7), the model revealed that increased gaze transition entropy was associated with lower pointing error ($\beta = -0.31$, 95% CrI = [$-0.48$, $-0.14$]). This corresponds to a 27% reduction in error ($\exp(-0.31) \approx 0.73$), suggesting that more dispersed gaze patterns foster flexible spatial representations.

Fixation duration also contributed to performance. Dwelling longer on buildings reduced pointing error ($\beta = -0.04$, 95% CrI = [$-0.06$, $-0.03$]), reflecting a 4% improvement per unit increase. Dwelling on agents showed a similar, albeit smaller, effect ($\beta = -0.03$, 95% CrI = [$-0.05$, $-0.01$]), translating to a 3% reduction. These findings support the value of sustained fixations in encoding task-relevant cues, though their effect was modest compared to entropy. Contextual features also influenced recall accuracy. Participants performed better in public buildings relative to residential ones ($\beta = -0.09$, 95% CrI = [$-0.14$, $-0.04$]), corresponding to a 9% error reduction. Finally, we examined the impact of agent

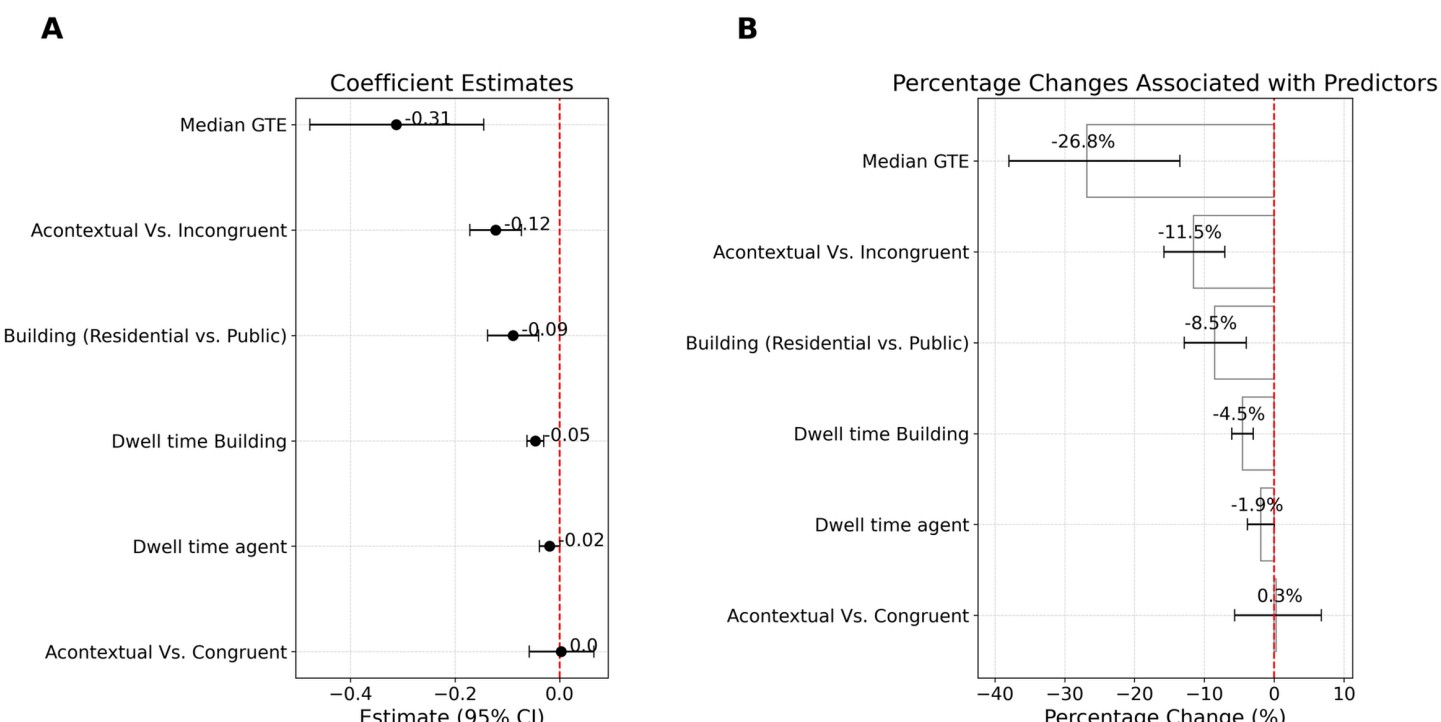

**Fig 7. Predicting spatial recall from gaze behavior.** (A) Regression surface showing relationship between GTE and pointing error, conditioned on agent dwell time. (B) Posterior estimates of fixed effects including gaze entropy, dwelling times, and contextual contrasts. 95% credible intervals shown.

congruency. Relative to the acontextual baseline, the congruent condition yielded a non-credible effect ($\beta = 0.01$, 95% CrI = [–0.06, 0.07]), while the incongruent condition reduced pointing error ($\beta = -0.12$, 95% CrI = [–0.17, –0.07]), amounting to a 11% improvement ($\exp(-0.12) \approx 0.89$). This suggests that incongruent agents, despite violating expectations enhanced memory.

In summary, these results highlight gaze entropy as a robust predictor of spatial recall accuracy. While prolonged fixations on agents and buildings also improved performance, entropy-driven exploration exerted the strongest effect. Furthermore, agent incongruency and public context facilitated better recall, reinforcing the importance of both attentional dynamics and environmental features in shaping spatial memory.

**Counterfactual analysis.** We assessed whether agent effects were mediated by post-encounter gaze transition entropy (GTE) using Bayesian parametric $g$-computation with a Beta–logit mediator model and a Gamma–log outcome model. Counterfactual predictions fixed dwelling covariates at their standardized means (0 on the $z$-scale) and were averaged over building type using empirical weights.

For congruent agents (vs. acontextual), the indirect effect via GTE was negative and credibly different from zero (NIE = $-0.167$, 95% CrI [–0.315, –0.054]; $RR_{NIE} = 0.996$ [0.993, 0.999]). The direct effect was uncertain and compatible with no material change (NDE = 0.273, 95% CrI [–2.660, 3.263]; $RR_{NDE} = 1.006$ [0.943, 1.071]), yielding a total effect near zero (TE = 0.106, 95% CrI [–2.823, 3.083]; $RR_{TE} = 1.002$ [0.939, 1.067]). Thus, this contrast is dominated by a small GTE-mediated reduction in error that is offset by an uncertain direct component, producing essentially no overall change.

For incongruent agents (vs. acontextual), the indirect effect via GTE was again small but reliable (NIE = $-0.177$, 95% CrI [–0.306, –0.073]; $RR_{NIE} = 0.996$ [0.993, 0.998]). In addition, there was a sizeable direct improvement not captured by GTE or dwelling times (NDE = $-5.174$, 95% CrI [–7.367, –3.013]; $RR_{NDE} = 0.888$ [0.844, 0.933]), producing a strong total reduction in error (TE = $-5.351$, 95% CrI [–7.530, –3.211]; $RR_{TE} = 0.884$ [0.840, 0.929]), i.e., 11% lower expected error. This indicates that, relative to acontextual scenes, a small GTE-mediated benefit combines with a sizeable direct pathway not captured by GTE or dwelling times.

Taken together, GTE provides a modest but consistent mediating pathway across contrasts, while incongruent agents additionally improve spatial memory via a substantial direct component. We therefore interpret GTE as one mechanism by which expectancy violations reorganize exploration in an encoding-relevant way, without claiming that it exhausts the social or motivational channels by which agents influence behavior.

# Discussion

This study investigated how human agents influenced visual exploration and learning within a controlled virtual reality city, focusing on how agent-context congruency modulates gaze behavior. The results revealed that agents shaped visual exploration patterns, especially when incongruent with their surroundings. Incongruent agents attracted longer dwelling time and increased gaze transitional entropy, promoting more dispersed and less predictable exploration patterns. This suggests that contextual mismatches between agents and their environment trigger attentional shifts beyond the immediate fixation, leading to more distributed scanning behavior. Crucially, gaze transitional entropy, not just experimentally manipulated factors or dwelling time, emerged as the most robust predictor of spatial learning performance. Higher entropy during exploration was associated with more accurate spatial recall. To examine the mechanism, we used counterfactual mediation with GTE as the mediator and absolute pointing error predicted on the original Gamma scale. The analysis indicated a modest GTE-mediated benefit in incongruent contexts together with a larger direct component not captured by GTE or dwelling times, whereas congruent contexts showed only a small mediated pathway and little net change. In practice, contextual mismatch reshaped exploration policies in a way that reliably improved performance, most clearly in incongruent scenes. In sum, the findings highlight the role of social-contextual information in shaping gaze behavior during spatial learning, demonstrating that incongruent agents enhance visual exploration and, consequently, support the acquisition of spatial knowledge.

While these findings offer new insights into how social agents modulate visual behavior and may inform theories of spatial learning and spatial learning, several limitations should be acknowledged. First, although participants were free to navigate the virtual environment, all objects and agents were non-animated, and the environment itself, while realistic, remained a simplified model of an urban space. This may limit the generalizability of the results to real-world navigation, where movement and environmental dynamics are known to influence attentional allocation [31,32]. However, introducing motion selectively was not feasible, as only contextual agents were equipped with objects or implied actions. Adding animation would have introduced an additional confound, undermining the internal validity of the congruency manipulation. To avoid this, a non-animated environment was employed, ensuring that observed differences in gaze behavior are most plausibly attributable to agent–context congruency. Future research could build on these findings by incorporating more dynamic and complex environments, allowing a deeper understanding of how social and contextual factors interact with gaze behavior under motion.

Second, while key visual properties such as agent size, gender, and skin color were controlled, and the distribution of graffiti was balanced across conditions, not all low-level visual features were fully standardized. Agent posture and object properties were left unconstrained to preserve ecological validity and the effectiveness of the congruency manipulation. Although gaze allocation is sensitive to perceptual features such as color or contrast [33], evidence from both experimental and computational studies indicates that gaze behavior in naturalistic tasks is primarily shaped by behavioral goals and semantic relevance [34,35]. This is especially relevant for entropy-based metrics, as gaze entropy reflects the interaction of bottom-up and top-down processes, with higher entropy often associated with cognitive control and goal-directed exploration [15,16]. Therefore, while low-level salience may have influenced gaze behavior to some extent, the observed modulation of gaze transition entropy by agent congruency likely reflects meaningful, task-driven exploration dynamics supporting spatial learning.

Agent-context congruency shaped gaze behavior along four distinct empirical dimensions. First, incongruent agents attracted longer fixation durations than both congruent and acontextual agents, suggesting that contextual violations elicited increased local attentional engagement, consistent with previous research showing that semantic inconsistencies prolong fixation times [36]. Second, incongruent agents were associated with higher gaze transition entropy, reflecting a shift from localized and predictable exploration to more distributed and flexible scanning patterns, aligning with findings that expectancy violations induce broader visual exploration [37]. Third, incongruent agents intensified the exploration-exploitation trade-off. Specifically, the negative relationship between dwelling time and entropy was stronger for incongruent agents, indicating that prolonged fixations on these agents were systematically followed by increased gaze dispersion across the environment. Finally, this reorganization of gaze patterns had a direct and robust effect on spatial learning. Gaze transition entropy emerged as the strongest predictor of spatial recall performance, surpassing the predictive value of dwelling time alone. More specifically, when context is congruent, the entropy pathway to better recall is modest and there is no consistent residual effect. When context is incongruent, entropy again contributes, but a larger direct route remains, indicating additional processes that enhance learning beyond changes in exploration policy. In sum, agent-context congruency systematically shaped participants' gaze behavior, accounting for their performance in the spatial memory task to moderate degree.

By violating contextual expectations, incongruent agents not only prolonged fixations on themselves but also promoted a transition from localized exploitation to more distributed and flexible exploration, as reflected in increased gaze transition entropy. This is consistent with the idea that gaze behavior is sensitive not only to the physical properties of stimuli but to their social and contextual significance [38,39]. In this study, social cues regulated visual attention and its distribution across space, redirecting participants' exploration. While previous studies have demonstrated that agents can guide attention toward semantically relevant locations or shape exploratory behaviour through social wayfinding cues [4,5], the present findings indicate that agents can induce a redistribution of visual attention when they violate environmental expectations.

This aligns with evidence that expectancy violations lead to extended search behavior and attentional shifts [40]. Specifically, contextual violations prolong fixations on the anomalous element [36] even when at odds with current task demands [37], and redistribute sampling across the scene [41]. However, the mnemonic impact of contextual violations is mixed. Reports range from better change detection or scene recall [42] to impairments [37,43]. In our data, incongruence strengthened the inverse dwelling–GTE relation, consistent with a two-stage adjustment process that begins with a brief examination of the anomaly followed by a broader exploration. Thus, anomaly-driven visual redistribution aids recall but does not fully explain it.

These findings contribute to the growing discussion on the role of humans as environmental facilitators during spatial learning [44,45]. Previous work has found that agents provide semantic structure, guide attention, and support memory formation during navigation. Specifically, human agents in an environment shape spatial learning through the semantic information they convey and the expectations they evoke, rather than functioning as static referential cues [5]. For example, socially relevant agents can influence navigation by providing wayfinding cues, shaping exploratory decisions, and supporting spatial perspective-taking [4,46]. Evidence from both real and virtual navigation contexts shows that social relevance enhances gaze allocation [47,48] and improves spatial perspective-taking compared to purely directional or non-social cues [49]. Alternative interpretations remain plausible. Elevated GTE could reflect greater interest, task engagement, or motivation rather than a restructured sampling policy, and contextual mismatch may co-vary with arousal or low-level salience. Although agents were visually identical and models adjusted for dwelling time with a hierarchical structure, the residual effect for incongruent scenes cautions against a uniquely social account. Future work should dissociate socialness from anomaly by comparing human agents with non-social violations matched on novelty and salience, and manipulate incentives and motivation. Finally, counterfactual mediation rests on the assumption of no unmeasured confounding; the modest but reliable GTE pathway is consistent with a mechanistic role for sequence structure, but it does not exclude complementary channels such as schema updating or distinctiveness. This distinction is critical, as it suggests that the benefits of social agents during navigation stem partially from the structural reorganization they induce in visual behavior during spatial exploration.

## Conclusion

In conclusion, spatial cognition is not driven solely by geometry or task demands but is shaped by contextual expectations, including the congruency of elements within a scene. In our study, incongruent human agents were associated with higher gaze transition entropy (GTE), a shift toward more diversified exploration, and improved spatial recall, whereas congruent agents produced only a small entropy-related pathway and little overall performance change. Although GTE elucidates one mechanism through which contextual anomalies enhance memory, the residual direct effect and the assumptions underlying our mediation model indicate that additional processes such as schema updating, distinctiveness, arousal, or motivational factors likely operate in parallel. Together, these findings suggest that even weak human cues, including those not directly tied to navigational goals, can scaffold learning by shaping how environments are sampled and encoded. Understanding how such signals modify sequence structure and uncertainty is essential for building ecologically valid accounts of spatial learning.

## Acknowledgments

The authors sincerely thank all those who supported this project. Special appreciation goes to Nora Maleki and Linus Tiemann for their contributions to the development of the VR city and the Pointing Tasks used in the experiments. They also thank Philipp Spaniol for his 3D artwork and for implementing differential level-of-detail loading for the agents within the scene. Lastly, we thank Melissa Sarria-Mosquera and Kaya Gärtner for their support in data collection.

## Author contributions

**Conceptualization:** Tracy Sánchez Pacheco, Debora Nolte, Sabine U König, Peter König.

**Data curation:** Tracy Sánchez Pacheco, Debora Nolte.

**Formal analysis:** Tracy Sánchez Pacheco.

**Funding acquisition:** Gordon Pipa, Peter König.

**Investigation:** Tracy Sánchez Pacheco.

**Resources:** Sabine U König, Gordon Pipa, Peter König.

**Supervision:** Sabine U König, Gordon Pipa, Peter König.

**Validation:** Tracy Sánchez Pacheco, Debora Nolte.

**Visualization:** Tracy Sánchez Pacheco.

**Writing – original draft:** Tracy Sánchez Pacheco, Sabine U König, Peter König.

**Writing – review & editing:** Tracy Sánchez Pacheco, Debora Nolte, Sabine U König, Gordon Pipa, Peter König.

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
