## [Decision Letter · Decision Letter 0]

24 Jul 2025

PCOMPBIOL-D-25-01068

Beyond the first glance: How human presence enhances visual entropy and promotes spatial learning

PLOS Computational Biology

Dear Dr. Sanchez Pacheco,

Thank you for submitting your manuscript to PLOS Computational Biology. After careful consideration, we feel that it has merit but does not fully meet PLOS Computational Biology's publication criteria as it currently stands. Therefore, we invite you to submit a revised version of the manuscript that addresses the points raised during the review process.

Please submit your revised manuscript within 60 days Sep 23 2025 11:59PM. If you will need more time than this to complete your revisions, please reply to this message or contact the journal office at ploscompbiol@plos.org. Please include the following items when submitting your revised manuscript:

We look forward to receiving your revised manuscript.

Kind regards,

Tarkeshwar Singh, Ph.D

Guest Editor

PLOS Computational Biology

Joseph Ayers

Section Editor

PLOS Computational Biology

**Journal Requirements:**

At this stage, the following Authors/Authors require contributions: Tracy Lorraine Sanchez Pacheco, Debora Nolte, Sabine U König, Gordon Pipa, and Peter König. Please ensure that the full contributions of each author are acknowledged in the "Add/Edit/Remove Authors" section of our submission form.

- ® on page: 4.

Potential Copyright Issues:

- Figures 1, 2, and 3. Please confirm whether you drew the images / clip-art within the figure panels by hand. If you did not draw the images, please provide (a) a link to the source of the images or icons and their license / terms of use; or (b) written permission from the copyright holder to publish the images or icons under our CC BY 4.0 license. Alternatively, you may replace the images with open source alternatives. See these open source resources you may use to replace images / clip-art:

**Reviewers' comments:**

Reviewer's Responses to Questions

**Comments to the Authors:**

Reviewer #1: Sánchez Pacheco and colleagues present a re-analysis of gaze and location-recall data from a free-viewing experiment in VR. In this experiment, participants explored a virtual city with different types of virtual human agents placed throughout the city that were either congruent, incongruent, or not related to the context they were presented in (given by the buildings behind them). In this re-analysis, the authors calculate, in addition to the more traditional fixation times, a measure of gaze transition entropy (GTE) and show that this measure not only differs between different types of agents, but was also the best predictor of pointing accuracy towards remembered locations.

This is a well-written manuscript that makes a compelling case for GTE as a measure of variability of exploration through gaze. The data analyses and modelling are convincing. I do have some questions about how clearly pure memory effects of incongruence can be dissociated from the effect of co-occurring changes in GTE and consequently, about GTE as a predictor of performance in the memory task. Other than that, I only have rather minor comments.

Please find my detailed comments below.

1. My one bigger issue concerns the untangling of the effects of incongruency's cognitive effects, and of increased gaze entropy as its behavioural consequence. The authors demonstrate that participants were better at remembering locations of incongruent agents over congruent or acontextual ones. As the hierarchical models show, both agent incongruence and GTE are predictive of pointing accuracy towards remembered locations, with GTE being clearly the strongest predictor. However, the authors then write: "This suggests that incongruent agents—despite violating expectations—may enhance memory through increased gaze dispersion or salience." (l.411)

There are several issues with this. (i) There is literature to suggest that incongruence may help memory under certain conditions (e.g., O’Sullivan, C. S., & Durso, F. T. (1984). Effect of schema-incongruent information on memory for stereotypical attributes. Journal of Personality and Social Psychology, 47(1), 55–70. -- not quite the same, of course, as location is a rather different property compared to what was used there, but the key point still stands of incongruence being helpful rather than detrimental), so this is not necessarily surprising. But more importantly, (ii) this suggests that incongruence affects memory performance only or at least primarily through altered gaze behaviour, but should the modelling not provide independent contributions of each predictor? Note that I am also missing more explicit formulations of the models in the paper, so perhaps these could help clarify. In any case, if the models do not provide this, then this would complicate the interpretation of other parts of the analyses, as dwell-time measures would also be correlated to other predictors. Point (ii) is hinted at in several other places including the abstract but, unless I am missing or misunderstanding something (which is entirely possible), not really supported by the analyses. I would ask the authors to clarify, and be precise in their interpretations.

2. I commend the authors for making their data and code available. However, while much of the raw data is relatively straightforward, the data would benefit from some description, especially given that there is not the same number of data files per participant as there are experimental blocks. It is also not clear to me why the code is stored separately.

3. l.136: Please describe the algorithm here, this is not something the reader should have to go to a different paper for.

4. Are the SDs for fixation duration across agent or across participant?

5. Why are acontextual agents not mentioned at all in ll.285-304?

6. Please explicitly specify the full hierarchical models.

Minor comments / typos:

7. l.203: State the default priors, please.

8. eq. 5, eq. 6: In the context of hierarchical modelling, using sigma is potentially confusing here.

9. Conditions are sometimes capitalised (e.g., l.227, figure caption for fig. 3), other times not (e.g., l.212, most instances in the main text). Please be consistent.

10. Fig. 3: It is almost impossible to tell in the resolution provided what object the agent is holding (knowing that Fig. 3b is congruent, I assume it is a sandwich?), which appears to be central information. Larger panels A-C could help, but also a more informative figure caption.

Reviewer #2: Letter to the authors is attached.

**Have the authors made all data and (if applicable) computational code underlying the findings in their manuscript fully available?**

Reviewer #1: Yes

Reviewer #2: Yes

PLOS authors have the option to publish the peer review history of their article (what does this mean?). If published, this will include your full peer review and any attached files.

Reviewer #1: No

Reviewer #2: **Yes:** Philipp Stark

**Figure resubmission:**
---

## [Decision Letter · Decision Letter 1]

24 Nov 2025

PCOMPBIOL-D-25-01068R1

Beyond the first glance: How human presence enhances visual entropy and promotes spatial learning

PLOS Computational Biology

Dear Dr. Sanchez Pacheco,

Thank you for submitting your manuscript to PLOS Computational Biology. After careful consideration, we feel that it has merit but as you will see one Reviewer still has some minor concerns. Therefore, we invite you to submit a revised version of the manuscript that addresses the points raised.

Please submit your revised manuscript within 30 days Jan 24 2026 11:59PM. If you will need more time than this to complete your revisions, please reply to this message or contact the journal office at ploscompbiol@plos.org. Please include the following items when submitting your revised manuscript:

We look forward to receiving your revised manuscript.

Kind regards,

Tarkeshwar Singh, Ph.D

Guest Editor

PLOS Computational Biology

Joseph Ayers

Section Editor

PLOS Computational Biology

**Reviewers' comments:**

Reviewer's Responses to Questions

**Comments to the Authors:**

Reviewer #1: The revised version is much improved. My concerns have been addressed, I have no further comments.

Reviewer #2: I would like to begin by sincerely thanking the authors for the substantial effort they have put into this revision. The manuscript has improved considerably, both in clarity and analytical depth. I also wish to acknowledge that my previous review was rather strict, and I apologize if it came across as overly critical. My intention was to support the strengthening of an already promising study, and I very much appreciate how thoroughly the authors have addressed the earlier concerns.

In its current form, I find the paper well-written, methodologically sound, and highly relevant. I do not believe that further substantial changes are needed.

There is, however, one point I would still like to raise for clarification:

Line 14–16: I would appreciate more clarification here. If I understand correctly, the sentence suggests that the presence of a human agent (as opposed to the absence of an agent) influences exploration, visual attention, and memory recall, according to Sánchez-Pacheco et al. (2025, https://doi.org/10.3389/frvir.2025.1497237).

However, in Sánchez-Pacheco et al. (2025), I only find the following:

• “However, the overall difference between the experiment types was not significant (β Experiment = −2.15, p = 0.50), suggesting that the presence of agents did not universally affect visit counts across all sessions.”

• Visual attention, proxied by dwell time, was not compared between agent and non-agent environments, but only between contextual and acontextual agents (as well as congruent and incongruent conditions).

• The same applies to spatial recall in the “Performance Hypothesis.”

Therefore, I am not sure the authors can make a statement about the effect of the presence of a human agent (versus no human agent) based on the available evidence.

If this is the case, it becomes difficult to argue that the human agent itself is responsible for the observed effects.

My guess is that this role: “Despite their ubiquity in real-world navigation, the role of fellow humans in shaping spatial knowledge formation remains underexplored” can not be explored by this study and is therefore somehow confusing in this context.

Maybe by just separating the points, this paragraph becomes clearer (just a suggestion):

(a) Semantic relevance to an environment (perception of spatial relationships)

(b) Human agents as amplifiers for spatial awareness and knowledge acquisition

Apart from this, I am very satisfied with the current version of the manuscript. The authors have done an excellent job refining both the theoretical framing and the empirical analyses, and the paper now reads clearly and convincingly. I believe it makes a valuable contribution to the field and would fully support its publication once the above minor clarification has been addressed.

**Have the authors made all data and (if applicable) computational code underlying the findings in their manuscript fully available?**

Reviewer #1: Yes

Reviewer #2: Yes

PLOS authors have the option to publish the peer review history of their article (what does this mean?). If published, this will include your full peer review and any attached files.

Reviewer #1: No

Reviewer #2: No

**Figure resubmission:**
---

## [Editor Report · Decision Letter 2]

21 Dec 2025

Dear Sanchez Pacheco,

In the previous round, Reviewer 2 had raised one minor concern about conclusions derived from the available evidence. In my opinion, you have addressed this concern satisfactorily. Therefore, I am pleased to inform you that your manuscript 'Beyond the first glance: How human presence enhances visual entropy and promotes spatial learning' has been provisionally accepted for publication in PLOS Computational Biology.

Best regards,

Tarkeshwar Singh, Ph.D

Guest Editor

PLOS Computational Biology

Marieke van Vugt

Section Editor

PLOS Computational Biology

---

## [Editor Report · Acceptance letter]

PCOMPBIOL-D-25-01068R2

Beyond the first glance: How human presence enhances visual entropy and promotes spatial learning

Dear Dr Sanchez Pacheco,

I am pleased to inform you that your manuscript has been formally accepted for publication in PLOS Computational Biology. Your manuscript is now with our production department and you will be notified of the publication date in due course.

With kind regards,

Zsofia Freund
